# Effect of the PSSMA Content on the Heat Transfer Performances of Polyurea Nano-Encapsulated Phase Change Materials

**DOI:** 10.3390/ma14123157

**Published:** 2021-06-08

**Authors:** Jun-Won Kook, Kiseob Hwang, Jun-Young Lee

**Affiliations:** Research Institute of Sustainable Manufacturing Systems, Intelligent Sustainable Materials R&D Group, Korea Institute of Industrial Technology, 89 Yangdaegiro-gil, Ipjang-myeon, Seobuk-gu, Cheonan-si 31056, Chungcheongnam-do, Korea; kukjw83@kitech.re.kr (J.-W.K.); ks_hwang@kitech.re.kr (K.H.)

**Keywords:** heat transfer performance, polyurea, nano-encapsulated, PSSMA content, phase behavior

## Abstract

Polyurea nano-encapsulated phase change materials (PUA-NEPCMs) were prepared from an *n*-octadecane core and through the formation of amide bonds by the reaction of toluene 2,4-diisocyanate and poly(4-styrenesulfonic acid-co-maleic acid) sodium salt (PSSMA), followed by the subsequent formation of a PUA shell using a miniemulsion system. The effects of the synthetic conditions on the thermal properties and encapsulat ion effect of the NEPCMs were systematically investigated. Differential scanning calorimetry (DSC) revealed that the melting enthalpy and encapsulation efficiency of the PUA-NEPCMs prepared under optimal conditions reached 123.00 J/g and 54.27%, respectively. Although previous results suggested that the introduction of PSSMA results in a reduced heat transfer performance for NEPCMs, DSC analysis of the prepared PUA-NEPCMs showed that increasing PSSMA contents enhanced the heat transfer performance due to a decrease in the degree of supercooling. Our results could therefore lead to further enhancements in the heat transfer performance of PUA-NEPCMs, in addition to expanding their field of application.

## 1. Introduction

The storage of latent heat using a phase change material (PCM) is the most widely known method of storing heat energy, and is one potential method for improving storage efficiencies. In general, PCMs absorb or release large amounts of energy in the form of latent heat during a phase transition. In addition, PCMs can store heat during the melting process and release it during the solidification process. The majority of PCMs tend to exhibit low thermal conductivity, and so the combination of PCMs with materials exhibiting high thermal conductivities has been examined to improve the thermal conductivities of PCMs. Indeed, such mixtures have also been found to enhance the thermal properties of the original material. In addition, microencapsulated PCMs (MEPCMs) have emerged as potential materials for improving the heat transfer rate owing to their large surface area-to-volume ratios. Moreover, nano-encapsulated PCMs (NEPCMs) have attracted increasing attention because of their larger areas of heat transfer compared to MEPCMs; this has been attributed to their increased surface area. Furthermore, nano-encapsulation prevents leakage of the PCM core, and facilitates the control of volume changes during the associated phase changes. NEPCMs can be broadly applied in textiles [1,2], buildings [3,4], solar thermal energy storage systems [5,6], waste heat recovery systems, and high-temperature thermal energy storage systems [7,8,9], among others.

Currently, a range of techniques exists for the encapsulation of PCMs. In the case of NEPCMs, the shell material plays an important role in determining both the heat transfer properties and the mechanical strength of the final material. In terms of the organic polymers employed as shell materials, PUF [10,11], PUA [12,13,14], PMMA [15,16], and PMF [17,18,19] have received attention because of their compatibility and ease of manufacture. However, NEPCMs and polymer-based MEPCMs tend to exhibit low thermal conductivities. To address this issue, materials exhibiting high thermal conductivities have been introduced into the polymer shells, including graphene [20], graphene oxide [21,22,23], carbon nanotubes [24], and nano-silver [25]. More specifically, Jiang et al. [26] synthesized MEPCMs based on a paraffin core and nano-Al_2_O_3_-doped poly(methyl methacrylate-co-methyl acrylate) as the shell. They found that the thermal conductivity increased upon increasing the nano-Al_2_O_3_ content, while the latent heat decreased.

One of the biggest issues arising from the encapsulation of PCMs is the supercooling phenomenon, where latent heat is released at a lower temperature as the crystallization temperature decreases. Supercooling is a significant disadvantage in energy storage applications because it allows latent heat to be released over a lower or wider temperature range. Numerous studies have therefore been conducted to reduce the supercooling temperature, wherein the most common method reported involves the addition of a nucleating agent, such as a high melting point PCM, or solid nanoparticles. However, this method also reduces the latent heat of the encapsulated PCM due to additive loading [27]. In recent years, research has been conducted to suppress supercooling by optimizing the configuration or structure of the shell. For example, Tang et al. [28] developed a less-supercooled microencapsulated material (Micro-18) by preparing a copolymer using ODMA-MAA as the shell. As a result, the starting crystallization temperature of Micro-18 was 4 °C lower than that of the original n-octadecane (OD). It was therefore confirmed that homogeneous nucleation can be mediated by shell-induced nucleation when the shell composition and structure are optimized without additional additives during nucleation. In some of the literature reviewed above, it can be seen that considerable efforts have been made to reduce the supercooling of MEPCMs. However, limited studies exist regarding the supercooling of NEPCMs, with the exception of a few cases where fillers with high thermal conductivities are added to the shell.

Thus, we herein report the introduction of a miniemulsion system to optimize the composition and structure of a NEPCM shell without the requirement for additional additives or emulsifiers during manufacture of the NEPCMs using OD as the core. An amide bond is formed by introducing toluene 2,4-diisocyanate (TDI) and poly(4-styrenesulfonic acid-co-maleic acid) sodium salt (PSSMA) as intermediates to allow the formation of PUA through polymerization at the PCM droplet interface in the physically generated nano-sized nucleus. In addition, the effect of varying the PSSMA content in the prepared PUA NEPCMs on the shell thickness is examined, and the sizes, crystallization temperatures, and heat transfer performance of the prepared PUA-NEPCMs are evaluated. The novelty of this study therefore lies in the fact that we report an increase in the heat transfer performance (supercooling degree) of PAU-NEPCMs by increasing the PSSMA content during the PUA polymerization process at the PCM droplet interface.

## 2. Materials and Methods

### 2.1. Materials

TDI, OD, and PSSMA (M_n_ = 20,000 g/mol) were purchased from Sigma Aldrich (St. Louis, MA, USA). Ethylene diamine (EDA) was obtained from Samchun Pure Chemicals (Pyeongtaek, South Korea). All chemicals were used as received without further purification. Deionized (DI) water was used for all experiments.

### 2.2. Methods

#### 2.2.1. Synthesis of PUA-NEPCMs Using a Miniemulsion System

To prepare the PUA-NEPCMs, OD (9 g) was mixed with TDI (3 g) to form an oil phase. For the aqueous phase, the amount of PSSMA was varied between 1.0 and 5.0 g, as desired. PSSMA was dissolved in water (130 mL) at 80–90 °C, and following stabilization, the combined solution was emulsified using a homogenizer (T25 basic ULTRA-TURAX, IKA, Staufen, Germany) at 50 °C and 9000–10,000 rpm for 10 min. After homogenization, the emulsion was prepared in a 300 mL four-necked double-jacketed glass reactor equipped with a mechanical stirrer, three inlets, and a thermostat. The reaction was performed at 600 rpm and 65 °C for 5 h. When the reaction was complete, the temperature of the formed latex was slowly reduced to room temperature to prevent the polymer shell from breaking due to shrinkage of the core.

#### 2.2.2. Characterization

The core-shell particle size and size distribution were measured by dynamic laser scattering (DLS, Zetasizer Nano S, Malvern Instruments, Worcestershire, UK). The morphology of the core-shell nanoparticles was observed using field-emission scanning electron microscopy (FE-SEM, JSM 6701F, JEOL, Tokyo, Japan) and transmission electron microscopy (TEM, JSM100CXⅡ, UHR, JEOL, Tokyo Japan). Thermogravimetric analysis (TGA, TGA Q50, TA Instruments, Newcastle, DE, USA) was used to examine the thermal degradation properties of the synthesized nanoparticles. TGA was performed from 30 to 600 °C at a heating rate of 10 °C/min. The capacity of latent heat storage $\left(\Delta H\right)$ was measured by differential scanning calorimetry (DSC, DSC Q10, TA instruments, Newcastle, DE, USA) under a nitrogen atmosphere. Heating and cooling was carried out between 0 and 150 °C at a rate of 10 °C/min.

## 3. Results and Discussion

### 3.1. Synthetic Route to the PUA-NEPCMs

Scheme 1 shows a schematic diagram of the interfacial condensation polymerization process, wherein OD (the PCM), TDI, and PSSMA (a polymeric surfactant) are subjected to a high shear. Due to the fact that it is impossible to manufacture core-shell nanoparticles using a general interfacial condensation polymerization approach, the application of a high shear using a homogenizer is required. As a miniemulsion, the PCM droplets are broken down to yield nano-sized particles using a physical method, and subsequent rapid stirring at 600 rpm using a mechanical stirrer allows the preparation of the core-shell nanoparticles.

Here, if the speed of the mechanical stirrer is reduced, the small nanodroplets aggregate within a short time, and the particle size gradually increases. Table 1 shows the compositions of the prepared PUA-NEPCMs and their corresponding sizes. The ratio of TDI to EDA was fixed at 1:3, and the PSSMA content was varied. It was found that the PUA-NEPCM particle size decreased as the content of PSSMA increased. Since TDI and PSSMA react at the PCM droplet interface to form an amide bond (Figure 1), EDA was introduced as a crosslinking agent, thereby resulting in the formation of PUA through polymerization at the PCM droplet interface, to yield the PUA-NEPCMs. TGA was then carried out to examine the thermal properties of the core and the shell. Appendix A shows the TGA curves of the PUA-NEPCMs prepared by varying the PSSMA content. As can be seen from the figure, the thermal decomposition temperature of OD was ~150 °C, while PUA and PSSMA exhibited thermal decomposition temperatures of 260–280 and 440–450 °C, respectively.

### 3.2. Morphological Analysis of the PUA-NEPCMs

To confirm the spherical shapes and structures of the PUA-NEPCMs, FE-SEM and TEM analyses were conducted. Figure 2a,c show the morphology of the PUA- NEPCM prepared using 2 g PSSMA, and Figure 2b,d show the morphology of the PUA-NEPCM prepared using 4 g PSSMA. As outlined in Table 1, upon increasing the PSSMA content from 2 to 4 g, the PUA-NEPCM size slightly decreased from ~195 to 180 nm, as determined by FE-SEM (Figure 2a,b). In addition, the distinct shape of the PUA-NEPCMs was revealed by TEM (Figure 2c,d). Indeed, morphological analysis of the PUA-NEPCMs confirmed the successful polymerization of PUA through the reaction of urea with TDI and EDA at the PCM droplet interface.

### 3.3. Phase Change Behaviors of the PUA-NEPCMs

As shown in Figure 3, the phase change enthalpies of the PUA-NEPCMs with various PSSMA contents were recorded. The onset temperatures for the heating and cooling cycles were ~27 °C. Generally, the phase transition temperature of a PCM can be changed by varying the chain length of the carbon structure. However, the latent heat capacity of OD did not change during melting and solidification since no chemical reaction between the PCM molecules took place during the melting and freezing cycles. The melting and freezing temperatures ($T_{m},\;T_{f})$ of the PUA-NEPCMs, along with their melting and freezing latent heats $\left(\Delta H_{m},\;\Delta H_{f}\right),$ were obtained from DSC analysis, as detailed in Table 2. The differences between the melting and freezing temperatures of the PUA-NEPCMs containing different amounts of PSSMA are also given in Table 2. Overall, it was found that as the PSSMA content increased, the melting and freezing latent heat capacities gradually decreased, as did the encapsulation efficiencies. The encapsulation efficiencies (E, %) of the PUA-NEPCMs were calculated using the following equation:(1)$$E=\frac{\Delta H_{m,\;NEPCM}+\Delta H_{f,\;NEPCM}}{\Delta H_{m,\;OD}+\Delta H_{f,\;OD}}\;\times \;100$$
where $\Delta H_{m,\;NEPCM}$ and $\Delta H_{f,\;NEPCM}$ are the enthalpies of phase change during the melting and freezing of the PUA-NEPCM, respectively, while $\Delta H_{m,\;OD}$ and $\Delta H_{f,\;OD}$ are the enthalpies of phase change during the melting and freezing of pure OD, respectively. Thus, as the PSSMA content increased, the encapsulation efficiency calculated by Equation (1) decreased from 54.27 to 34.99%. This was attributed to an increase in the shell thickness of the PUA-NEPCMs upon increasing amount of PSSMA used in the reaction of TDI, PSSMA, and EDA at the PCM droplet interface. It should be noted that the surfactant does not participate in the PUA-NEPCM polymerization process, and that the emulsion is stabilized by the Pickering effect after droplet formation using a homogenizer.

More specifically, a Pickering emulsion is an emulsion that is stabilized by solid particles that adsorb onto the interface between the two phases. If oil and water are mixed and small oil droplets are formed and dispersed throughout the water, the droplets will eventually coalesce to decrease the amount of energy in the system. However, if solid particles are added to the mixture, they will bind to the surface of the interface and prevent droplet coalescence, thereby rendering the emulsion more stable [29].

### 3.4. Heat Transfer Performance of the PUA-NEPCMs Upon Variation in the PSSMA Content

We employed the values of $T_{m}$ and $T_{f}$ obtained by DSC analysis to measure the heat transfer performance of the PUA-NEPCMs. The size of the prepared PUA-NEPCMs and the phase change enthalpies during melting and freezing are related to the phase change behaviors and thermal properties of the NEPCMs. Thus, DSC melting and freezing peaks of NEPCMs prepared using varying amounts of PSSMA were obtained, and the results are summarized in Table 2.

The supercooling temperature of OD can therefore be evaluated using Equation (2) [30]:(2)$$\Delta T=T_{m}-T_{f}$$
where $T_{m}$ and $T_{f}$ are the melting and freezing temperatures, respectively. Generally, the melting and freezing temperatures can be represented by the melting and freezing peaks obtained from the DSC trace, respectively.

To evaluate the heat transfer performance of the prepared PUA-NEPCMs, their supercooling temperatures were calculated based on the data presented in Table 2, and the results are given in Figure 4. More specifically, Figure 4 shows the supercooling temperatures and melting latent heats for the various NEPCMs prepared using different PSSMA contents. As can be seen from the figure, an increase in the PSSMA content resulted in a reduction in the latent heat of the PUA-NEPCMs and a lower supercooling temperature. This was attributed to either the larger surface area or the greater number of polar hydroxyl groups on the PUA-NEPCMs upon increasing the PSSMA content, which in turn minimized the degree of supercooling. In our case, TDI and PSSMA react at the PCM droplet interface to form an amide bond, and as the PSSMA content increases, the number of hydroxyl groups at the PUA-NEPCM interface also increases. As a result, the supercooling temperature decreases with increasing PSSMA content. These results therefore indicate that the heat transfer performance of PUA-NEPCMs can be adjusted by variation in the PSSMA content, thereby suggesting a feasible approach to minimizing the supercooling temperatures of PUA-NEPCMs, and broadening their prospective applications.

## 4. Conclusions

We herein reported the preparation of nano-encapsulated phase change materials (NEPCMs) using an n-octadecane (OD) core, the formation of an amide bond through the reaction of toluene 2,4-diisocyanate (TDI) and poly(4-styrenesulfonic acid-co-maleic acid) sodium salt (PSSMA), and the formation of polyurea (PUA) using a miniemulsion system at the OD droplet interface. Since the direct use of PCMs has limitations in terms of their applicability and compatibility, we selected PUA as a modifier polymer due to its good thermal conductivity. In terms of the effect of the PSSMA content, upon increasing the loading, it was found that the PUA-NEPCMs decreased in size slightly, and the core-shell structure could be clearly confirmed. In addition, upon the introduction of PSSMA, the maximum melting and freezing latent heats obtained were 123.00 and 128.90 J/g, respectively, while the maximum encapsulation efficiency was 54.27%. Subsequently, the relationship between the PSSMA content and the supercooling temperatures (heat transfer performance) of the PUA-NEPCMs was investigated, as was that between the PSSMA content and the melting latent heats of the PUA-NEPCMs. It was found that as the PSSMA content was increased from 1.0 to 5.0 g, the heat transfer performance of the PUA-NEPCMs decreased from 18.01 to 11.47 °C. Our results therefore indicate that the PUA-NEPCMs developed herein constitute a new material that can reduce the supercooling temperature of PCMs to produce highly valuable heat dissipating materials for application in the fields of heat control fabrics, solar cells, and electronic devices. Overall, our results indicated the successful design of a material with improved heat transfer performance that could be tuned according to the PSSMA content. Although various known methods exist that can reduce the supercooling degree of PCMs, no previous studies have shown this effect clearly. Future work in this area will therefore be focused on the development of a new heat-resistant material through reducing the supercooling temperature of such PCMs [31].

## Data Availability

The data presented in this study are available upon request from the corresponding author.

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
