# Peer review of "Effect of the PSSMA Content on the Heat Transfer Performances of Polyurea Nano-Encapsulated Phase Change Materials"

_materials, 2021, doi:10.3390/ma14123157_

Round 1

Reviewer 1 Report

The paper is scientifically good and that it is clearly written.

 A few points concerning the paper to help the authors:

1) Design the details of a simple model that can help general readers to understand the processes involved into the investigated properties

2) Underline better potential applications of the materials discussed

3) Compare the present results with already known methods and materials in the literature

4) Expand the conclusions in order to highlight the importance of the research here presented

The authors could design a simple model that can rationalise the presented experimental results.

Author Response

Answer sheets

Materials (Manuscript ID: materials-1215985)

“Effect of the PSSMA Content on the Heat Transfer Performances of Polyurea Nano-encapsulated Phase Change Materials” by Jun-Won Kook et el.

Author’s Reply to the Report of Reviewer 1:

Thank you for comments and suggestions.

First of all, in the process of modifying the manuscript, we discussed with the authors and revised the title as shown in “Effect of the PSSMA Content on the Heat Transfer Performances of Polyurea Nano-encapsulated Phase Change Materials.”

The paper is scientifically good and that it is clearly written.

A few points concerning the paper to help the authors:

1) Design the details of a simple model that can help general readers to understand the processes involved into the investigated properties

    In this report, when the content of PSSMA in PUA-NEPCM was increased from 1 to 5 g, the ∆T of PUA-NEPCM decreased by DSC analysis. In order to increase hear transfer performance, heat transfer in the shell must be good. So, a polymer called PUA was selected and designed to be confirmed by introducing a supercooling degree. Therefore, we think it can present the possibility of new research that can increase the heat transfer performance of PUA-NEPCM.

2) Underline better potential applications of the materials discussed

Thank you for this suggestion. We have now revised the introduction to make reference to selected solar or high-temperature thermal energy storage (recovery) as potential applications for our materials. In this context, current research is focusing on the use of microencapsulated PCMs for combination with inorganic materials to produce products suitable for solar heat or high-temperature thermal energy storage applications, and on the development of materials that possess a sufficient durability at high temperatures. Thus, in contrast to the original PCM, our material exhibits a large surface area and is successful in reducing the supercooling temperature; we have now emphasized those points and the potential applications of our material in the revised manuscript (lines 39–43).

3) Compare the present results with already known methods and materials in the literature

Thank you for this suggestion. We would like to point out that a number of appropriate literature references have been made in the article. For example, equation (1) in Section 3.3 is presented with reference to citation [13], while equation (2) and Figure 4 in Section 3.4 were referenced to citation [30], and the values of Tm, Tf, and ∆T given in Table 2 were obtained from Table 1 of citation [14]. Thus, all equations and data presented in this paper are verified by the corresponding literature reference.

We would also like to point out here that materials introduced in previous studies have ranged from inorganic filler materials to oxide-based and acrylate-based monomers, and these were mainly composed of MEPCMs. In contrast, we report the preparation of PUA-NEPCMs via an interfacial condensation polymerization approach, wherein PSSMA was introduced during the PUA polymerization process to increase the heat transfer performance of the final material.

4) Expand the conclusions in order to highlight the importance of the research here presented

Thank you for this suggestion. The conclusions section has now been expanded and rewritten to highlight the importance of our research. In particular, we added a reference [31] related to the application of heat-resisting materials in PCM to the end of the conclusion (lines 252–261).

The authors could design a simple model that can rationalise the presented experimental results.

Reviewer 2 Report

In Figure 2a and 2b, few large particles and other numerous uniform small particles are observed, what are these uniform small particles?

Compared to the reported PCM or commercialized product PCM, which kind of improvement were observed in the newly synthesized Polyurea Nano-encapsulated Phase Change Materials in this work?

Author Response

Answer sheets

Materials (Manuscript ID: materials-1215985)

“Effect of the PSSMA Content on the Heat Transfer Performances of Polyurea Nano-encapsulated Phase Change Materials” by Jun-Won Kook et el.

Author’s Reply to the Report of Reviewer 2:

Thank you for comments and suggestions.

First of all, in the process of modifying the manuscript, we discussed with the authors and revised the title as shown in “Effect of the PSSMA Content on the Heat Transfer Performances of Polyurea Nano-encapsulated Phase Change Materials.”

In Figure 2a and 2b, few large particles and other numerous uniform small particles are observed, what are these uniform small particles?

Thank you for this question. The many uniform small particles observed in Figures 2a and 2b are secondary particles. More specifically, upon the polymerization of PUA at the PCM droplet interface, PUA-NEPCM forms, although some single PUA particles are also present, therefore accounting for the observed uniform small particles.

Compared to the reported PCM or commercialized product PCM, which kind of improvement were observed in the newly synthesized Polyurea Nano-encapsulated Phase Change Materials in this work?

In the case of previously reported PCMs and commercially available PCMs, many restrictions exist in terms of their potential applications due to issues related to supercooling. We therefore considered that the encapsulation of PCM with a polymer would give a material suitable for application as a heat storage or heat transfer material. This study proposes a method to reduce the supercooling degree of PCM by controlling the PSSMA content during the PUA polymerization process at the PCM droplet interface. Using DSC analysis of the samples containing different PSSMA contents, it was observed that heat transfer performance (i.e., the supercooling degree) was successfully improved compared to previously reported (or commercialized) PCMs.

Reviewer 3 Report

Manuscript titled: Heat Transfer Performance of Polyurea Nano-encapsulated Phase Change Materials based on PSSMA contents, has been reviewed and the following comments have been proposed:

1- In abstract, innovation and necessity of this research are not clearly presented.

2- In the introduction, it is better to compare the samples of past research works with the proposed method in this research in order to highlight the capability and innovation of this research.

3- Methodology is general and simple. It is better to present the details and all the steps by graphs, scientific equations and relations with the existing standards and criteria. It is better to validate the proposed method.

4- The conclusion is not complete. The details of the different parts of the results and the advantages and disadvantages existing method should be clearly concluded based on the main and applied goals. Conclusion details of each section should be classified.

Author Response

Answer sheets

Materials (Manuscript ID: materials-1215985)

 "Effect of the PSSMA Content on the Heat Transfer Performances of Polyurea Nano-encapsulated Phase Change Materials” by Jun-Won Kook et el.

Author’s Reply to the Report of Reviewer 3:

Manuscript titled: Heat Transfer Performance of Polyurea Nano-encapsulated Phase Change Materials based on PSSMA contents, has been reviewed and the following comments have been proposed:

Thank you for comments and suggestions.

First of all, in the process of modifying the manuscript, we discussed with the authors and revised the title as shown in “Effect of the PSSMA Content on the Heat Transfer Performances of Polyurea Nano-encapsulated Phase Change Materials.”

1- In abstract, innovation and necessity of this research are not clearly presented.

Thank you for pointing out this omission. The innovation and necessity have now been clearly indicated in the abstract, as indicated in blue text (lines 18-22).

2- In the introduction, it is better to compare the samples of past research works with the proposed method in this research in order to highlight the capability and innovation of this research.

Thank you for this comment. However, we believe that appropriate literature comparisons have already been made in the introduction. For example, in lines 62-74, we reviewed the contents of past research, while in lines 74-83, we added some details that emphasize the capability and innovation of this research compared to previous works.

3- Methodology is general and simple. It is better to present the details and all the steps by graphs, scientific equations and relations with the existing standards and criteria. It is better to validate the proposed method.

Thank you for this suggestion. We would like to point out that a number of appropriate literature references have been made in the article. For example, equation (1) in Section 3.3 is presented with reference to citation [13], while equation (2) and Figure 4 in Section 3.4 were referenced to citation [30], and the values of Tm, T f, and ∆T given in Table 2 were obtained from Table 1 of citation [14]. Thus, all equations and data presented in this paper are verified by the corresponding literature reference.

4- The conclusion is not complete. The details of the different parts of the results and the advantages and disadvantages existing method should be clearly concluded based on the main and applied goals. Conclusion details of each section should be classified.

Thank you for this suggestion. The conclusions section has now been completely revised and rewritten for emphasis and to include the requested details. For example, conclusions related to the morphologies of the PUA-NEPCMs have been added (lines 243-245), and additional information regarding the scope of application of the PUA-NEPCMs and the possibility of further research in this area have been added. In particular, we added a reference [31] related to the application of heat-resisting materials in PCM to the end of the conclusion (lines 252-261).

Round 2

Reviewer 3 Report

The authors have responsively improved the manuscripts based on the comments raised in the first review round. The manuscript now looks qualified enough to be considered for publication.